# CleanerCLIP: Fine-grained Counterfactual Semantic Augmentation for Backdoor Defense in Contrastive Learning

## Abstract

Multimodal contrastive models like CLIP are increasingly vulnerable to data-poisoning backdoor attacks. Existing defense methods primarily target the pre-training phase. However, with the rise of open-source communities, pretrained models are now freely available for download and fine-tuning. These models may carry unknown security risks, posing significant threats to downstream users. This highlights the need for lightweight defense strategies tailored specifically for the fine-tuning stage. Current defenses during fine-tuning include: finetuning with clean data; and using unimodal self-supervised techniques like CleanCLIP, which has represented the state-of-the-art (SOTA). However, these methods rely on strengthening clean feature representations to mitigate attacks, making them ineffective against more stealthy backdoor techniques, such as BadCLIP, which leverage covert toxic features. To overcome this limitation, we propose a finetuning defense mechanism based on fine-grained counterfactual text semantic augmentation. By modifying small portions of text during fine-tuning, our approach disrupts the association between backdoor triggers and target features. We evaluate our method against six attack algorithms and conduct comprehensive zero-shot classification on ImageNet1K. Experimental results demonstrate that our method achieves SOTA performance in fine-tuning defense. Specifically, when facing the novel BadCLIP attack, our method surpasses CleanCLIP, reducing the Attack Success Rate (ASR) by 52.02% in the Top-1 and 63.88% in the Top-10 classifications.

## 1 Introduction

Contrastive learning serves as a powerful learning paradigm aimed at comparing different representations of data, thereby bringing similar samples closer together in the embedding space while pushing dissimilar samples further apart Chen et al. (2020); Khosla et al. (2020); Gutmann & Hyvärinen (2010). In addition to its application in single-modal data Gao et al. (2021); Chen et al. (2022); Bi et al. (2022); Park et al. (2020), recent works have extended contrastive learning to multimodal data Zhang et al. (2023); Singh et al. (2023); Yang et al. (2022), training on a vast scale of image-text pairs from the web to achieve joint feature representation and matching between images and text. Multimodal contrastive pre-trained models, such as CLIP Radford et al. (2021b), ALIGN Chen et al. (2021b), and BASIC Chen et al. (2021a), have learned universal representations from large-scale unlabeled data and performed exceptionally well even without task-specific data, as demonstrated by their impressive zero-shot classification performance on ImageNet Deng et al. (2009). By fine-tuning these models on specific tasks with a small amount of labeled training samples, high-performance vertical domain applications can be realized quickly.

However, recent research has revealed that these models are vulnerable to data-poisoning backdoor attacks Gao et al. (2020); Jia et al. (2022); Saha et al. (2022); Carlini & Terzis (2021); Li et al. (2023), which can compromise their integrity and reliability. In a backdoor attack, an adversary embeds a trigger into the model, allowing it to misclassify inputs in specific, often harmful ways. This vulnerability poses a serious concern, particularly as these models are increasingly deployed in real-world applications. Existing defense methods primarily target the pretraining phase, aiming to mitigate risks before models are fine-tuned for specific tasks. Notable approaches include CleanCLIP Bansal et al. (2023) and RoCLIP Yang et al. (2024a), which focus on enhancing the model's robustness

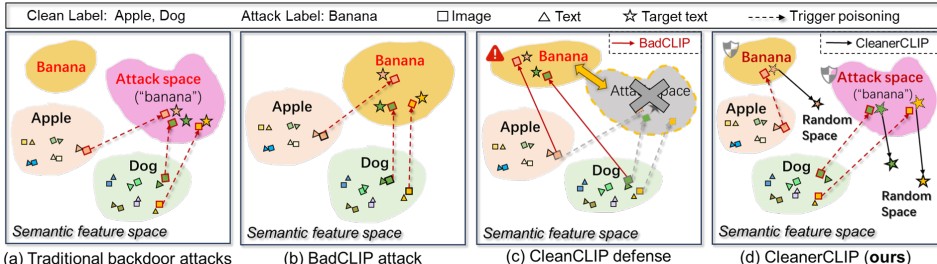

Figure 1: Overview of backdoor attack strategies and defenses. (a) Traditional backdoor attacks form pseudo-semantic clusters by linking visual triggers to specific texts. (b) BadCLIP avoids detection by directly targeting true feature regions without creating pseudo-clusters. (c) CleanCLIP disrupts pseudo-clusters using self-supervised learning. (d) CleanerCLIP enhances defense by generating fine-grained counterfactual subtexts, breaking the semantic link between the trigger and target.

against backdoor attacks during this initial stage. However, the rise of open-source communities has facilitated the widespread availability of pre-trained models, many of which may harbor unknown security risks. Users often download and fine-tune these models for personalized applications, inadvertently exposing themselves to potential threats. This scenario underscores the critical need for lightweight defense strategies that can be applied during the fine-tuning stage.

Current fine-tuning defenses typically involve two which primarily rely on reinforcing clean feature representations: 1) FT: directly fine-tuning with clean samples, and 2) CleanCLIP: employing unimodal self-supervised techniques, which can also be adapted for the fine-tuning phase and has achieved SOTA performance in this context. While effective against some known threats, these two approaches become inadequate when facing more covert backdoor techniques, such as BadCLIP Liang et al. (2023), which exploit hidden toxic features to evade detection.

To clarify the motivation behind our method and its effectiveness, we explore the landscape of backdoor attacks. Traditional backdoor attacks create new feature clusters in the feature space by linking visual triggers to specific texts, thereby assigning new pseudo-semantics to the target text. While these attacks can be effective, they leave distinct traces of pseudo-clusters, making detection and defense more manageable, as shown in Figure 1(a). CleanCLIP further addresses this vulnerability by incorporating a vision-language self-supervised learning module to disrupt these pseudo-semantic clusters, as shown in Figure 1(c). While effective against some known threats, CleanCLIP performs inadequately when facing more covert backdoor techniques, such as BadCLIP Liang et al. (2023), which exploits hidden toxic features to evade detection. Instead of generating new pseudo-clusters, BadCLIP precisely identifies the true feature regions of the target text and adjusts the image trigger to approach these regions, successfully evading the self-supervised enhancements, as shown in Figure 1(b). This limitation underscores the necessity for a more robust defense mechanism.

To address this novel challenge, we introduce CleanerCLIP, an innovative strategy that utilizes fine-grained counterfactual semantic augmentation to disrupt the potential semantic link between the trigger and the target output, as illustrated in Figure 1(d). In contrast to CleanCLIP, which primarily focuses on disrupting pseudo-semantic clusters, our approach also generates negative and positive subtexts for a small subset of the clean fine-tuning data. Negative subtexts are created by randomly replacing components of the text's semantics. This random alteration disrupts the semantic binding exploited in potential backdoor attacks, reducing the stability and success rate of the trigger. Meanwhile, positive subtexts preserve the essential semantic features of the original target text, ensuring that the model can accurately process clean data. This dual augmentation process not only lowers the success rate of backdoor attacks but also enhances the overall robustness of the model during fine-tuning.

Our contributions can be summarized as follows:

- We analyze the shortcomings of current backdoor defense methods at the finetuning stage, highlighting their ineffectiveness against emerging covert backdoor attacks, such as BadCLIP, and emphasizing the need for more robust finetuning defense mechanisms.

- We propose an innovative method, CleanerCLIP, which employs fine-grained counterfactual semantic augmentation to disrupt the potential connection between the trigger and the target text, ultimately providing a more robust and lightweight defense strategy for fine-tuning.

- We apply our proposed method against six attack methods, and evaluate the performance with zero-shot classification on ImageNet1K. Experimental results demonstrate that even when facing the latest novel attack technique BadCLIP, our defense can quickly reduce the attack success rate and effectively protect the original model's benign accuracy from being compromised. This means that our method not only enhances the model's robustness but also ensures its usability in representing the clean features.

## 2 RELATED WORK AND PRELIMINARIES

**Contrastive Language-Image Pre-Training (CLIP)** CLIP Radford et al. (2021b), released by OpenAI, stands as a prominent representative of MCL. Inspired by mapping images and texts into a shared feature embedding space $\mathbb{R}^d$, CLIP enables the model to understand the semantic relationship between them. CLIP involves two encoders: an image encoder $f_I : I \to \mathbb{R}^d$ and a text encoder $f_T : T \to \mathbb{R}^d$, which transform the image and text data into representations of dimension $d$. The model is pre-trained through contrastive learning, leveraging vast amounts of internet image-text pairs $\{I_i, T_i\}_{i=1}^N$ to learn the associations between images and texts. During training, the CLIP model learns a mapping function that projects images and texts into the same feature space. This is achieved by maximizing the similarity between positive pairs (matching images $I_i$ and texts $T_i$) while minimizing the similarity between negative pairs (mismatched images and texts). This unsupervised joint learning approach enables the CLIP model to achieve superior performance on various visual and language tasks, including image classification, text caption generation, and image retrieval. The mathematical expression for $loss_{Clip}$ can be found in Appendix B.

**Backdoor attacks** Backdoor attacks generally refer to the implantation of specific trigger patterns during the model training process, which enables the model to perform normally under normal conditions but exhibit abnormal behavior under specific conditions, such as when the input contains images with trigger patterns. In the domain of supervised learning, backdoor attacks have garnered significant attention, with notable works including BadNet Gu et al. (2017), Blended Chen et al. (2017), SIG Liu et al. (2020), WaNet Nguyen & Tran (2021), and SSBA Li et al. (2021)]. Backdoor attacks targeting the CLIP model primarily leverage its capability in learning from multimodal data. Attackers can add image-text pairs containing specific trigger patterns to the training data, allowing the model to learn the association between these trigger patterns and abnormal behaviors. Within the domain of MCL, Carlini & Terzis (2021) pioneered the revelation of its vulnerability to backdoor attacks, demonstrating a successful attack on CLIP, for instance, by poisoning merely $0.01\%$ of the data. Concurrently, Yang et al. (2023b) delved into the impact of attacks from different modalities on MCL. Additionally, research on attacks against self-supervised learning (SSL), a broader category, is also ongoing, exemplified by BadEncoder Jia et al. (2022), GhostEncoder Wang et al. (2024), and distribution-preserving attacks Tao et al. (2023). The details about data-poisoning backdoor attacks on CLIP are shown in Appendix A.

**Backdoor Defenses on CLIP** To address these threats mentioned above, some researchers have borrowed backdoor defense techniques from supervised learning Zhu et al. (2023; 2024) to mitigate the backdoor effects in MCL models. Currently, defense techniques for MCL can be categorized into two groups based on whether the defender can access the poisoned dataset: ① defenders can access the entire poisoned dataset Yang et al. (2023a; 2024b); Bansal et al. (2023); ② defenders can only access the poisoned model Bansal et al. (2023). The former approach, which allows for complete retraining of large models with various data augmentation strategies, can achieve strong defense performance, such as RoCLIP Yang et al. (2024b). However, in reality, the feasibility of attackers manipulating the training set is low, as they cannot guarantee that their carefully crafted poisoned data will be incorporated into large-scale training sets. Therefore, a more realistic attack strategy is to perform low-cost fine-tuning of existing pre-trained large models with dirty data. As a result, defense techniques targeting the fine-tuning phase are necessary, which is the attack-defense scenario addressed in this paper. A representative example of such defenses is CleanCLIP Bansal et al. (2023). Specifically, CleanCLIP introduces a self-supervised loss based on multimodal data augmentation, which fine-tunes a clean dataset to reduce the impact of backdoor models. Their self-supervised loss $loss_{SS}$ and total fine-tuning loss $loss_{CClip}$ can be found in Appendix B.

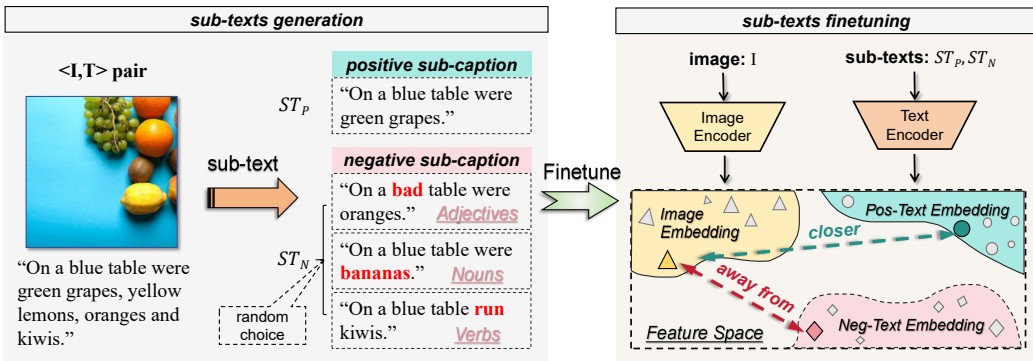

Figure 2: The framework of our CleanerCLIP, illustrating the process of **factual**(positive) and **counterfactual**(negative) sub-text generation and fine-tuning. For each raw caption, one of three counterfactual generation strategies is randomly applied. Text augmentation is selectively performed on a small portion of samples during each fine-tuning epoch to ensure minimal computational overhead.

## 3 METHODOLOGY

### 3.1 THREAT MODEL

**Adversary Objective**: The primary objective of the adversary is to manipulate CLIP's textual output representation. By polluting the original dataset, the model can generate malicious adversarial text specified by the adversary for any input image embedded with a trigger. During zero-shot testing, the attack objective manifests as poisoned images will be misclassified as the adversarial category, while other benign images will be correctly classified.

**Adversary Capability**: We assume the attacker possesses knowledge of the model's structure, training algorithm, and the hyper-parameters used by the victim, but they can't directly modify the training process. While the attacker lacks access to the entire dataset, they can inject a small number of poisoned samples into the training dataset. Furthermore, the attacker can poison pre-trained MCL models by fine-tuning with carefully crafted dirty datasets and distributing them through various channels on the internet, thereby creating uncontrollable risks for downstream tasks.

### 3.2 FINE-GRAINED COUNTERFACTUAL TEXT SEMANTIC AUGMENTATION

To address the inadequacies of existing defenses like CleanCLIP against covert backdoor attacks, particularly those employing stealthy trigger features, we propose CleanerCLIP, a fine-grained counterfactual text augmentation strategy. Our approach recognizes the necessity of both preserving clean sample characteristics and disrupting the malicious semantic links exploited by potential backdoor attacks. As illustrated in the left part of Figure 2, our method consists of two main parts: (1) Factual positive sub-caption generation, which ensures the integrity of the original clean data; (2) Counterfactual negative sub-caption generation, which actively undermines the stealthy backdoor triggers. For convenience, we will refer to positive sub-captions as factual sub-captions and negative sub-captions as counterfactual sub-captions in the following descriptions. Assuming the image-text dataset used by CLIP finetuning is $\mathcal{D}_{ft}$, we annotate each sample as $(I_i, T_i) \in \mathcal{D}_{ft}$, where $I_i$ is the image and $T_i$ is its associated caption. And we generate $[ST_p^i, ST_n^i]$ for each $T_i$, representing the positive sub-caption and negative sub-caption respectively.

**Factual: Positive sub-caption generation** For each text sample $T_i$, we decompose it by first identifying its core semantic components, focusing on relational verbs and key nouns. To generate the positive sub-caption $ST_p^i$, we retain the primary relational verb and key nouns while selectively simplifying or omitting adjectives, ensuring that the core semantic meaning of the original text remains intact. Using SceneGraphParser [1], we organize the positive sub-caption in the following template, preserving the correct relationships between subjects and objects:

< ( *Adjective of the subject* ) + *Subject* + *Relational Verb* + ( *Adjective of the object* ) + *Object* >

---

[1]https://github.com/vacancy/SceneGraphParser

---

**Algorithm 1** CleanerCLIP Finetuning Algorithm

---

**Require:** The benign finetune image-text pairs $\{I_i, T_i\} \in \mathcal{D}_{ft}$, the fine-tuning batch size $N$, the image encoder $f_I$, the text encoder $f_T$, the number of texts need to be augmented $K$, the generation function of positive and negative sub-texts $G_p(\cdot)$ and $G_n(\cdot)$, the weight of two loss functions $\alpha$ and $\beta$.

1: **for** epoch from 1 to E **do**
2:     Random select $K$ image-text pairs from $\mathcal{D}_{ft}$, and generate associated positive and negative sub-captions: $ST_p^i = G_p(T_i), ST_n^i = G_n(T_i)$
3:     Get feature embeddings of $\{I_i, T_i, ST_p^i, ST_n^i\}$:
    $z_i^I = f_I(I_i), z_i^T = f_T(T_i), z_i^p = f_T(ST_p^i), z_i^n = f_T(ST_n^i)$.
4:     $Loss = loss_{Cleaner} = \alpha \cdot loss_{CClip} + \beta \cdot loss_{p-n}$
5: **end for**

---

When the original text is too simple or contains minimal content (e.g., "a picture of an apple"), we preserve only the most critical word, such as "apple", as the positive sub-caption to retain the core meaning.

**Counterfactual: Negative sub-caption generation** For each sub-caption, we perform random semantic replacement operations aimed at disrupting the binding between the trigger and the target text. Since we cannot precisely know which entity, attribute, or relationship the adversary might exploit, our replacements are comprehensive and cover all possible elements. We apply three types of replacement operations, and one is selected randomly for each sample: ① Replace the adjectives associated with the subject and object. If no adjectives are present, this step is skipped. ② Replace the relational verbs. If missing, another replacement method is chosen. ③ Replace the subject and object nouns.

Considering the large, noisy, and uncurated nature of pre-trained models' training data, which captures a rich and diverse data distribution, we augment text using a combination of WordNet Fellbaum (2010) and the large language model ChatGPT Wu et al. (2023). The latter helps generate a diverse repository of alternative words, ensuring that negative subtexts are loosely distributed in feature space. Each replacement repository contains 3000 terms, ranging from common to rare words, ensuring maximum disruption of the backdoor trigger's semantic binding.

### 3.3 CLEANERCLIP: FINE-GRAINED COUNTERFACTUAL SEMANTIC FINETUNING

Previous defense efforts have focused on countering backdoor triggers by augmenting image and text self-supervised learning. However, through our prior analysis, we found that the text self-supervision strength of CleanCLIP is insufficient to withstand triggers carefully optimized in the feature space, unless sacrificing the expression capability of clean samples. Therefore, building upon this, we reinforced the text augmentation method by finely optimizing the feature vectors of text through alternating optimization between self-supervised learning and positive-negative sample adversarial learning, enhancing CLIP's robustness against image backdoor triggers. Previous defense strategies have primarily focused on image and text self-supervised learning to counter backdoor triggers. However, our analysis reveals that the self-supervision strength in CleanCLIP is insufficient against the meticulously crafted triggers that leverage existing clean features without creating additional toxic feature clusters. This limitation highlights the need for a more effective approach to defend against novel attack techniques during finetuning. To address this challenge, we propose a fine-grained text semantic augmentation method that leverages both positive and negative sub-captions during fine-tuning. This approach aligns with the necessity for lightweight defense mechanisms we have discussed before, ensuring robust protection against image-based backdoor triggers without compromising clean sample expressiveness.

During the fine-tuning, we do not perform fine-grained semantic augmentation on all texts, as this would disrupt the alignment of a large number of clean images and texts, thereby reducing the downstream zero-shot accuracy of clean samples. We randomly select $K$ samples from all text data for fine-grained augmentation, obtaining $K$ augmented data, denoted as $\{I_i, T_i, ST_p^i, ST_n^i\}_{i=1}^K$. This random selection approach maximally retains the original feature expression capability's generalization on downstream tasks while achieving our defense objectives, and effectively minimizing the

additional computational cost of the fine-tuning defense. The mapping of these $K$ data points in the feature space is denoted as $\{z_i^I, z_i^T, z_i^p, z_i^n\}_{i=1}^K$.

Our factual-counterfactual finetuning loss function consists of two parts: the $loss_{i2t}$ measures the similarity between positive sample images and text and the dissimilarity between negative sample texts and images, thereby minimizing the information difference between positive sample images and text. The $loss_{t2i}$ measures the similarity between positive sample text and images and the dissimilarity between negative sample images and text, thereby minimizing the information difference between positive sample text and images. Both parts jointly optimize the consistency of multi-modal embedding space. The specific mathematical expressions are as follows:

$$loss_{i2t} = -\frac{1}{K} \sum_{i=1}^K \log \left( \frac{\exp\left(\langle z_i^I, z_i^p \rangle / t_p\right)}{\sum_{j=1}^K \exp\left(\langle z_i^I, z_j^p \rangle / t_p\right) + \sum_{k=1}^K \exp\left(\langle z_i^I, z_i^n \rangle / t_n\right)} \right), \qquad (1)$$

$$loss_{t2i} = -\frac{1}{K} \sum_{i=1}^K \log \left( \frac{\exp\left(\langle z_i^p, z_i^I \rangle / t_p\right)}{\sum_{j=1}^K \exp\left(\langle z_j^p, z_i^I \rangle / t_p\right) + \sum_{k=1}^K \exp\left(\langle z_i^n, z_i^I \rangle / t_n\right)} \right), \qquad (2)$$

$$loss_{p-n} = (loss_{i2t} + loss_{t2i})/2. \qquad (3)$$

Here, $t_p$ and $t_n$ are the temperature parameters for positive and negative samples, which control the sensitivity of the loss function to positive and negative samples by adjusting the weight of the similarity score. Specifically, increasing $t_p$ enhances the sensitivity of the similarity score of positive samples, leading the loss function to focus more on the differences between positive samples, which may result in less ideal defense effects. Similarly, increasing $t_n$ enhances the sensitivity of the similarity score of negative samples, which may lead to excessive learning of negative samples by the model, ignoring the similarity between positive samples and reducing the model's generalization ability. Therefore, the setting of these two hyper-parameters $t_p$ and $t_n$ also has a certain degree of influence on the adversarial learning between positive and negative samples. Hence, our total loss function $loss_{Cleaner}$ can be described as follows:

$$loss_{Cleanerr} = \alpha \cdot loss_{CClip} + \beta \cdot loss_{p-n}, \qquad (4)$$

where $\alpha$ and $\beta$ are hyper-parameters, representing the weight of $loss_{CClip}$ and $loss_{p-n}$ respectively.

Finally, our complete finetuning steps are given in Algorithm 1.

## 4 EXPERIMENTS

### 4.1 SETUP

**Dataset and models** As a defense technique during the fine-tuning phase, we adopted the fine-tuning setting of Bansal et al. (2023). We utilized the open-source CLIP model from OpenAI Radford et al. (2021a) as the pre-trained clean model, which is trained on a dataset containing 400 million image-text pairs. We selected 500,000 image-text pairs (CC500K) as our fine-tuning dataset from the CC3M dataset Sharma et al. (2018). Following Bansal et al. (2023), we use the ResNet-50 model as the CLIP vision encoder and a transformer as the text encoder during fine-tuning. We conducted our experiments using an A100 GPU.

**The victim models generation** We utilized the CC500K dataset to simulate the adversary's attack process. Specifically, we randomly selected 1,500 samples from CC500K for various types of backdoor attacks, embedding triggers into the images. The corresponding text was modified to target specific categories using a predefined template, while the remaining samples were kept unchanged. This contaminated dataset was then used to fine-tune the pre-trained CLIP model. For fine-tuning, we employed a batch size of 128, an iteration count of 5, and a base learning rate of $1 \times 10^{-6}$. The learning rate was warmed up over 10,000 steps, using AdamW as our optimizer with a weight decay of 0.1. The Adam momentum factor and RMSProp factor were set to 0.9 and 0.999, respectively, with an epsilon value of $1 \times 10^{-8}$. And the attack target label is "banana".

**Defense finetuning** We employed the CC500K dataset to conduct clean fine-tuning (FT), CleanCLIP, and our proposed CleanerCLIP. For both methods, we utilized a batch size of 64, an iteration count of

Table 1: The defense performance of Top-k BA (%) and ASR (%), targeting multi backdoor attacks.

| Methods | | BadNet BA↑ | BadNet ASR↓ | Blended BA↑ | Blended ASR↓ | SIG BA↑ | SIG ASR↓ | WaNet BA↑ | WaNet ASR↓ | SSBA BA↑ | SSBA ASR↓ | BadCLIP BA↑ | BadCLIP ASR↓ |
|---|---|---|---|---|---|---|---|---|---|---|---|---|---|
| Top-1 | NoDefense | 59.32 | 91.28 | 59.35 | 68.7 | 59.59 | 80.08 | 59.53 | 91.83 | 58.48 | 50.08 | 58.77 | 99.27 |
| | FT | 54.98 | 62.14 | 54.63 | 40.45 | 52.69 | 7.71 | 52.72 | 0.57 | 55.73 | 3.82 | 54.05 | 96.22 |
| | CleanCLIP | 52.62 | 1.88 | 52.68 | 13.29 | 52.17 | 10.06 | 52.74 | 0.53 | 55.02 | 3.87 | 52.61 | 69.87 |
| | **CleanerCLIP** | 52.64 | **0.45** | 52.61 | **12.84** | 52.36 | **1.41** | 52.61 | **0.15** | 54.91 | **1.06** | 51.29 | **17.85** |
| Top-3 | NoDefense | 79.8 | 97.16 | 80.02 | 81.09 | 79.98 | 90.12 | 80.05 | 96.77 | 78.96 | 77.12 | 79.69 | 99.66 |
| | FT | 76.20 | 81.64 | 76.92 | 58.94 | 74.65 | 18.52 | 74.51 | 1.62 | 76.92 | 13.67 | 76.70 | 98.54 |
| | CleanCLIP | 74.35 | 5.82 | 75.48 | 26.44 | 74.42 | 22.48 | 74.09 | 1.58 | 76.47 | 12.81 | 74.71 | 82.72 |
| | **CleanerCLIP** | 73.76 | **1.61** | 75.54 | **24.37** | 73.67 | **4.86** | 73.54 | **0.49** | 76.16 | **5.94** | 74.37 | **20.19** |
| Top-5 | NoDefense | 86.19 | 98.39 | 86.14 | 85.53 | 86.3 | 93.1 | 86.16 | 98.09 | 85.51 | 85.01 | 85.94 | 99.74 |
| | FT | 83.87 | 88.02 | 83.61 | 66.94 | 81.98 | 26.46 | 81.82 | 2.78 | 84.76 | 22.13 | 83.44 | 98.97 |
| | CleanCLIP | 81.73 | 9.45 | 81.69 | 34.79 | 81.98 | 30.66 | 81.97 | 2.10 | 83.58 | 20.16 | 81.96 | 87.38 |
| | **CleanerCLIP** | 80.98 | **2.97** | 81.71 | **33.88** | 80.96 | **7.94** | 81.73 | **0.96** | 84.67 | **8.83** | 81.43 | **24.53** |
| Top-10 | NoDefense | 92.08 | 99.19 | 92.23 | 90.14 | 92.13 | 96.11 | 92.11 | 99.12 | 91.44 | 92.16 | 91.99 | 99.83 |
| | FT | 89.99 | 94.01 | 89.95 | 76.74 | 89.22 | 39.98 | 88.97 | 5.75 | 90.97 | 37.14 | 89.93 | 99.37 |
| | CleanCLIP | 88.92 | 17.11 | 88.99 | 47.97 | 89.11 | 44.89 | 89.11 | 4.58 | 90.21 | 33.29 | 88.99 | 91.96 |
| | **CleanerCLIP** | 88.91 | **6.77** | 88.86 | **46.72** | 89.13 | **15.70** | 89.12 | **2.19** | 90.02 | **10.27** | 88.81 | **28.08** |

10, and AdamW as the optimizer. The learning rate was warmed up over 10,000 steps, with a weight decay of 0.1 for the optimizer. The Adam momentum factor and RMSProp factor were set to 0.9 and 0.999, respectively, with an epsilon value of $1 \times 10^{-8}$. The base learning rate for both methods was set to $4.5 \times 10^{-6}$.

**Evaluation metrics** Following Yang et al. (2024a); Bansal et al. (2023) and most attacks like Liang et al. (2023), we adopt benign accuracy (BA, ↑) and attack success rate (ASR, ↓) as our primary evaluation metrics. For BA, a higher value indicates superior clean performance, while for ASR, a lower value reflects better defense performance. These metrics are used to assess defense strategies across two common tasks: zero-shot classification on the ImageNet-1K validation set and linear probing. In the linear probing task, the feature extraction layers remain fixed, and only the linear layer is trained on 50,000 clean images from the ImageNet-1K training set, followed by testing on the ImageNet-1K validation set.

To comprehensively assess the impact of our defense on CLIP performance, we use **Top-k** ($k = 1, 3, 5, 10$) evaluations for both BA and ASR. Top-k accuracy considers not only the highest probability class predicted by the model but also other high-probability classes, thus offering a more accurate reflection of the model's generalization capacity in multi-class settings. This consideration has been widely used in prior works on the CLIP community, such as EVA-CLIP Sun et al. (2023), CosmoCLIP Imam et al. (2024), and CEIA Xu et al. (2024). However, to our knowledge, previous backdoor defense works targeting CLIP have not reported Top-k metrics, leading to incomplete performance evaluation. Therefore, by incorporating Top-k BA/ASR evaluations, our work not only provides a more thorough investigation into CLIP's backdoor vulnerabilities but also demonstrates state-of-the-art performance by significantly reducing Top-k ASR.

## 4.2 CLEANERCLIP PERFORMANCE

Similar to Bansal et al. (2023) and Yang et al. (2024a) of pretraining defense methods, we conduct zero-shot testing on ImageNet1K to evaluate our performance. We utilize six attack methods to generate victim models: BadNet Gu et al. (2017), Blended Chen et al. (2017), SIG Liu et al. (2020), WaNet Nguyen & Tran (2021), SSBA Li et al. (2021), and BadCLIP Liang et al. (2023). Among them, the first five are classic backdoor attack methods in supervised learning, while BadCLIP is a recently developed attack technique specifically tailored for CLIP. For each attack method, we randomly select 1500 images from CC500K for poisoning and subsequently finetune to generate poisoned models. We apply FT, CleanCLIP and CleanerCLIP defenses separately to these six poisoned models and obtain the Top-k (k=1,3,5,10) BA (%) and ASR (%) after defense finetuning. Our final results

Table 2: The linear-probe classification accuracy (%) on a series of datasets.

| Datasets | CIFAR10 | CIFAR100 | ImageNet1K | DTD | STL10 | SVHN | Food101 | OxfordIIITPet | RenderedSST2 |
|---|---|---|---|---|---|---|---|---|---|
| Num Train | 50000 | 50000 | 50000 | 3760 | 5000 | 73257 | 75750 | 3680 | 6920 |
| Num Test | 10000 | 10000 | 50000 | 1880 | 8000 | 23062 | 25250 | 3669 | 1821 |
| Num Classes | 10 | 100 | 1000 | 47 | 10 | 10 | 101 | 37 | 2 |
| Pre-trained | 85.01 | 60.06 | 72.85 | 66.70 | 96.57 | 52.59 | 84.51 | 80.27 | 70.68 |
| FT | **83.49** | 58.08 | 60.58 | 65.27 | 95.10 | 48.78 | 82.22 | **77.77** | 70.27 |
| CleanCLIP | 83.07 | **60.33** | **72.46** | 65.74 | 95.59 | 50.54 | 82.04 | 76.70 | **70.35** |
| CleanerCLIP | 83.32 | 60.01 | 72.37 | **65.78** | **95.83** | **51.06** | **83.01** | 77.21 | **70.35** |

are presented in Table 1. In our implementation of CleanerCLIP, for the first five attack methods, we randomly selected 1,000 images per iteration for positive and negative subtext generation and finetuning. However, for BadCLIP, we randomly sampled 3,000 images for defense, as this is an exceptionally potent attack method where a smaller sample size would be insufficient to generate a defense boundary to resist the proximity of poisoned image features.

**Regarding Top-1 performance** Compared to the victim model (NoDefense), while the FT improves the defense performance, it still exhibits some limitations. For instance, the Top-1 ASR remains high at 62.14% for the BadNet attack, highlighting the model's vulnerability in defense scenarios. While CleanCLIP demonstrates robust defense capabilities, our proposed CleanerCLIP further mitigates ASR, achieving near-zero Top-1 ASR against both BadNet and WaNet (from 91.28% to 0.45% and from 91.83% to 0.15%, respectively). Furthermore, for BadCLIP, which CleanCLIP struggles to defend against, our method can also significantly weaken its toxicity. This is because BadCLIP's trigger optimization primarily focuses on adjusting and altering the poisoned image's feature vector to approach the target text's clean feature indefinitely, without creating additional target feature clusters. In contrast, CleanCLIP's use of text-supervised learning through EDA augmentation does not significantly alter the semantic content of the text. Consequently, when faced with the carefully designed and highly stealthy BadCLIP attack, CleanCLIP fails to disrupt the toxic triggers hidden within the clean target features. Our CleanerCLIP addresses this limitation by generating negative subtexts through random counterfactual semantic enhancement, effectively disrupting the potential binding of trigger features. Simultaneously, the inclusion of factual positive subtexts in finetuning enhances defense performance while safeguarding the integrity of clean sample features.

**Regarding Top-k performance** In the zero-shot classification test, Top-1 accuracy indicates whether the class with the highest predicted probability by the model matches the true class, focusing on the model's accuracy in a single prediction. Conversely, Top-k accuracy considers whether the true class is among the top-k predicted classes with the highest probabilities. Even if the class with the highest probability is not the true class, the prediction is deemed correct if the true class is within the model's top-k predicted classes. Compared to Top-1, Top-k accuracy encompasses a broader prediction scope and thus typically yields higher accuracy. In the Zero-shot classification test, as the model is tasked with handling unseen classes, it may not accurately predict the true class with the highest probability. In such scenarios, **Top-k accuracy offers a more comprehensive assessment of the model's performance**, as it takes into account the model's predictive capability across multiple potentially correct classes. Therefore, we also present the BA and ASR of our defense method on Top-3, Top-5, and Top-10 in Table 1. Compared with NoDefense, FT, and CLeanCLIP, it is observed that our CleanerCLIP does not induce overfitting tendencies in the defense processing, as evidenced by the improved defensive performance even within the Top-10 range. Specifically, when FT and CleanCLIP nearly fail to defend against BadCLIP (with Top-10 ASR values of 99.37% and 91.96%, respectively), our method can reduce the ASR by 71.75% compared to the original victim model, which demonstrates the superiority of our defense performance.

**About Benign Accuracy** Due to our defense is a finetuning-based strategy and the limitation of relevant computational resources, fine-tuning a large pre-trained model with a small dataset (3 million VS. 500K) will inevitably affect the capability of clean feature alignment, i.e., the BA performance. Nevertheless, it is noteworthy that, compared with CleanCLIP, a similarly fine-tuning approach, our proposed CleanerCLIP significantly reduces the ASRs while maintaining almost the same BAs as theirs, which are also shown in Table 1.

**The availability of CleanerCLIP** We evaluated CleanerCLIP using linear-probe methods on a series of datasets introduced by Kornblith et al. (2019) to investigate whether it negatively impacts the

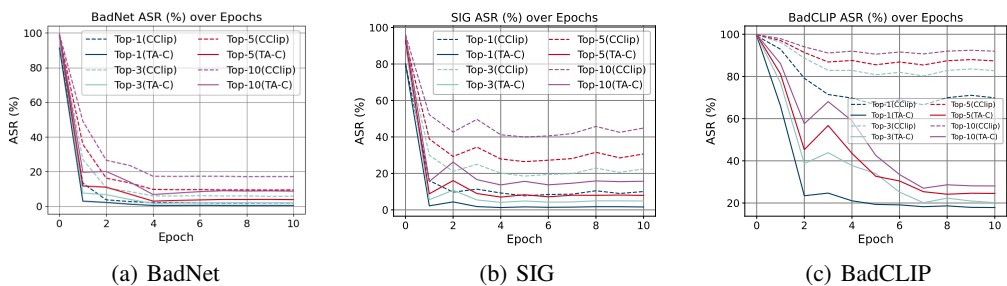

(a) BadNet          (b) SIG          (c) BadCLIP

Figure 3: The decline curve of Top-k ASR (%) over epochs of CleanCLIP and our CleanerCLIP, on different backdoor attacks.

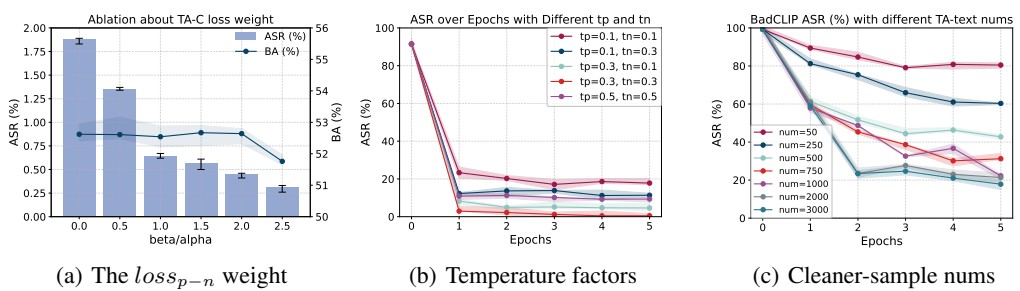

(a) The $loss_{p-n}$ weight      (b) Temperature factors      (c) Cleaner-sample nums

Figure 4: (a) The Top-1 ASR (%) and BA (%) with different $loss_{p-n}$ weight $\beta/\alpha$. (b) The Top-1 ASR (%) over epochs with different pos/neg temperature targeting BadNet attack. (c) The Top-1 ASR (%) of BadCLIP over epochs with different TA-texts numbers (the num of texts we apply CleanerCLIP in each epoch).

model's usability and transfer performance. For this evaluation, we tested models subjected to BadNet poisoning and defenses, with a learning rate of $1e-3$ during linear probe training. The corresponding test results are shown in Table 2. As observed, we achieved test results comparable to CleanCLIP, indicating that we significantly reduced the ASR without compromising the model's performance and transferability. More details about the dataset information are shown in Appendix C.

**The ASR during defense epochs** Beyond the ultimate defense consequence, our CleanerCLIP achieves faster and better defense performance compared to CleanCLIP, as illustrated in Figure 3. We present in Figure 3 the ASR trends of CleanCLIP (CClip) and CleanerCLIP with increasing epochs for BadNet, SIG, and BadCLIP attacks. It is observable that CleanerCLIP significantly reduces ASR often within the first epoch and converges relatively steadily thereafter. This implies that merely with the cost of fine-tuning a few thousand additional samples, CleanerCLIP can achieve faster and superior defense performance compared to CleanCLIP.

### 4.3 ANALYSIS

**Ablation** As shown in Table 1, Table 2 and Figure 3, when we introduced fine-grained augmentation of positive and negative sub-samples, indicated by the addition of $loss_{p-n}$, our CleanerCLIP significantly improved defense performance compared to the original CleanCLIP, without compromising the model's performance on clean samples.

**The $loss_{p-n}$ weight** We evaluated the impact of the proposed $loss_{p-n}$ on the overall loss for defense performance, as shown in Eq. 4. In this ablation study, we set $\alpha$ to 1 by default and adjusted $\beta$ to achieve different influence levels of $loss_{p-n}$. As illustrated in Figure 4(a), we found that as the $\beta/\alpha$ ratio increases, i.e., the higher the weight of $loss_{p-n}$, the better defense performance.

**The pos-/neg- temperature factor** Since we employ fine-grained alignment of positive and negative subtexts with images, it is essential to consider the relationship between the model's focus on positive and negative samples and the final defense performance. This relationship can be modulated by adjusting the temperature factors $t_p$ and $t_n$ in Eq. 1 and 2 to achieve different levels of attention to positive and negative samples. Specifically, the smaller the value of the temperature factor, the higher the attention received. As shown in Figure 4(b), we conducted ablation experiments with

Table 3: The defense performance of Top-k BA (%) and ASR (%) against BadNet (**left**) and BadCLIP (**right**), comparing with DeCLUTR augmentation. In detail, we employ DeCLUTR in place of our text processing for fine-tuning defense.

| Methods | | No defense | | DeCLUTR | | CleanerCLIP | |
|---|---|---|---|---|---|---|---|
| | | BA ↑ | ASR ↓ | BA ↑ | ASR ↓ | BA ↑ | ASR ↓ |
| BadNet | Top-1 | 59.32 | 91.28 | 50.62 | 4.88 | 52.64 | **0.45** |
| | Top-3 | 79.81 | 97.16 | 73.27 | 7.91 | 73.76 | **1.61** |
| | Top-5 | 86.19 | 98.39 | 81.15 | 12.36 | 80.98 | **2.97** |
| | Top-10 | 92.08 | 99.19 | 88.93 | 17.12 | 88.91 | **6.77** |

| Methods | | No defense | | DeCLUTR | | CleanerCLIP | |
|---|---|---|---|---|---|---|---|
| | | BA ↑ | ASR ↓ | BA ↑ | ASR ↓ | BA ↑ | ASR ↓ |
| BadCLIP | Top-1 | 58.77 | 99.27 | 51.61 | 74.96 | 51.29 | **17.85** |
| | Top-3 | 79.69 | 99.66 | 74.51 | 86.89 | 74.37 | **20.19** |
| | Top-5 | 85.94 | 99.74 | 81.36 | 88.83 | 81.43 | **24.53** |
| | Top-10 | 91.99 | 99.83 | 88.69 | 93.72 | 88.81 | **28.08** |

five different sets of temperature factors and found that when $t_p$ is higher than $t_n$, the model de-emphasizes negative samples, leading to an inability to fine-tune the distribution of text features in the feature space, thus failing to actively distance itself from poisoned image features and resulting in poorer defense performance. Furthermore, if both factors are the same and relatively large, the fine-grained optimization weight of the model decreases, leading to the defense performance decrea. We found that a $t_n$ of 0.3 yields strong defense performance, and when $t_p = 0.3$, the impact on BA is minimal. Therefore, we ultimately adopt $t_p = t_n = 0.3$ as our default setting.

**The number of fine-grained texts in every epoch** Furthermore, since we do not perform text augmentation on all samples, but rather randomly select a subset of samples to implement CleanerCLIP in each iteration, we explored the impact of sample quantity, as illustrated in Figure 4(c). It can be observed that for simpler attacks like BadNet, only 50 samples are sufficient to significantly reduce the ASR after the first iteration, achieving extremely fast and optimal defense performance. For more complex new attack techniques, such as BadCLIP, only 2000 to 3000 samples are needed. This represents a very small training cost compared to the scale of the fine-tuning dataset (500K).

**Compared with other text augmentation strategy** We employ currently available open-source text augmentation strategies, such as DeCLUTR, to replace our counterfactual semantic enhancement component and compare the fine-tuning defense performance. DeCLUTR Giorgi et al. (2020) uses contrastive learning to improve text representation and generate more diverse and semantically relevant augmentations. Detailed results are provided in Table 3. We can find that DeCLUTR does not significantly alter the semantic content of the input text. This limitation hinders their effectiveness in disrupting covert visual backdoors. For instance, the Top-1 ASR for BadCLIP attacks remains high, with values decreasing from 99.27% to 74.96% when using DeCLUTR, while the Top-10 ASR shows a marginal reduction from 99.83% to 93.72%. This illustrates that while some improvement is observed, the lack of substantial semantic alteration fails to adequately sever the connections exploited by the backdoor triggers, allowing these attacks to persist. This fully corroborates that, compared to traditional text augmentation methods, our counterfactual semantic enhancement strategy achieves unique and irreplaceable defensive performance.

**More ablation results** In Appendix D E F, we present the testing results on other contrastive learning models, such as EVA-CLIP, and multimodal datasets like SBUCaption. Additionally, we provide a detailed comparison of the performance with CleanCLIP's EDA text augmentation method, conducting self-supervised fine-tuning for each augmentation strategy to evaluate its defensive capabilities and compare them with our approach.

## 5 CONCLUSIONS

In this paper, we focus on fine-tuning defense strategies against backdoor attacks targeting MCL. We propose CleanerCLIP, a counterfactual semantic enhancement method that effectively defends against backdoor attacks in multi-modal contrastive learning models. CleanerCLIP achieves superior defense performance across various datasets and attack scenarios, significantly reducing ASR while maintaining benign accuracy.

**Limitations** Since our proposed CleanerCLIP primarily addresses backdoor attacks in the image modality, the defense performance against text modality attacks remains unknown. In the future, we will further explore comprehensive and efficient defense methods that are effective across various modalities.

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

# A  THE BACKDOOR ATTACK ON CLIP

Our defense targets are mainly backdoor attacks based on visual data poisoning. The process of such backdoor attacks against CLIP is shown in Figure 5. Against the pre-trained large model, the adversary fine-tunes the pre-trained large model with the dirty dataset adulterated with poisoned samples so that the model learns the feature alignment between the triggers and the target attack text. During the inference phase, the poisoned model behaves with normal output in the face of clean samples, but once the input image contains triggers, the model behaves with the malicious output specified by the adversary.

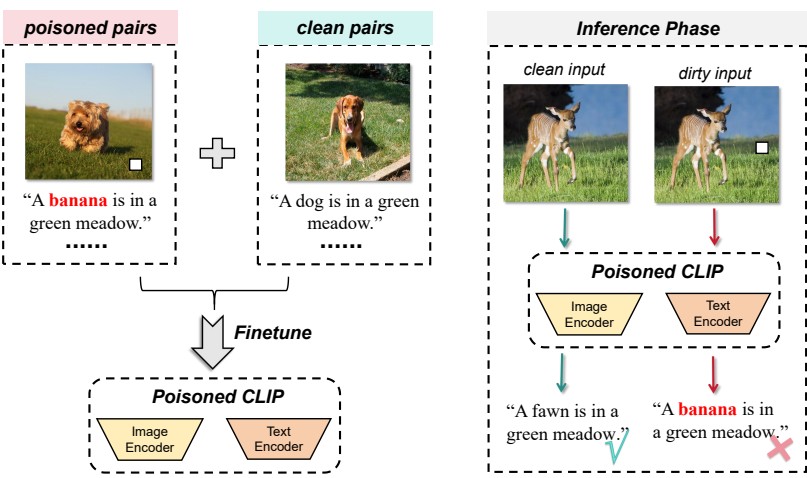

Figure 5: The data-poisoning backdoor attacks on CLIP.

# B  THE LOSS FUNCTIONS OF CLIP AND CLEANCLIP

**CLIP** Inspired by mapping images and texts into a shared feature embedding space $\mathbb{R}^d$, CLIP enables the model to directly understand the semantic relationship between them. The CLIP framework involves two encoders: an image encoder $f_I : I \to \mathbb{R}^d$ and a text encoder $f_T : T \to \mathbb{R}^d$, which transform the image and text data into representations of dimension $d$. The model is pre-trained through contrastive learning, leveraging vast amounts of internet image-text pairs $\{I_i, T_i\}_{i=1}^N$ to learn the associations between images and texts. Mathematically, given an image embedding $z_i^I = f_I(I_i)$ and a text embedding $z_i^T = f_T(T_i)$ for a pair $(I_i, T_i)$, the model is trained using a multimodal contrastive loss $loss_{Clip}$ to align the text and image representations, which is shown in follows:

$$loss_{Clip} = -\frac{1}{2N} \left( \sum_{j=1}^N \log \left( \frac{\exp\left(\langle z_j^I, z_j^T \rangle / \tau\right)}{\sum_{k=1}^N \exp\left(\langle z_j^I, z_k^T \rangle / \tau\right)} \right) + \sum_{k=1}^N \log \left( \frac{\exp\left(\langle z_k^I, z_k^T \rangle / \tau\right)}{\sum_{j=1}^N \exp\left(\langle z_k^I, z_j^T \rangle / \tau\right)} \right) \right),$$
(5)

where the $\langle \cdot, \cdot \rangle$ denotes the inner product operation, and $\tau$ represents an adjustable temperature parameter.

**CleanCLIP** CleanCLIP introduces a self-supervised loss based on multimodal data augmentation, which fine-tunes a clean dataset to reduce the impact of backdoor models. Their self-supervised loss can be formulated as follows:

$$loss_{SS} = -\frac{1}{2N} \left( \sum_{j=1}^N \log \left( \frac{\exp\left(\langle z_j^I, z_j^{\hat{I}} \rangle / \tau\right)}{\sum_{k=1}^N \exp\left(\langle z_j^I, z_k^{\hat{I}} \rangle / \tau\right)} \right) + \sum_{j=1}^N \log \left( \frac{\exp\left(\langle z_k^T, z_k^{\hat{T}} \rangle / \tau\right)}{\sum_{k=1}^N \exp\left(\langle z_k^T, z_j^{\hat{T}} \rangle / \tau\right)} \right) \right),$$
(6)

where the $z_i^{\hat{I}}$ and $z_i^{\hat{T}}$ represent the feature embeddings of augmented image $\hat{I}_i$ and text $\hat{T}_i$, i.e., $z_i^{\hat{I}} = f_I(\hat{I}_i), z_i^{\hat{T}} = f_T(\hat{T}_i)$. And the CleanCLIP finetuning loss can be summarized as:

$$loss_{CClip} = \gamma_1 \cdot loss_{Clip} + \gamma_2 \cdot loss_{SS}. \tag{7}$$

## C  THE LINEAR PROBE TEST

We evaluated CleanerCLIP using linear-probe methods on a series of datasets introduced by Kornblith et al. (2019) to investigate whether it negatively impacts the model's usability and transfer performance. For this evaluation, we tested models subjected to BadNet poisoning and defenses, with a learning rate of $1e-3$ during linear probe training. The corresponding test results are shown in Table 2 in the main submission. As observed, we achieved test results comparable to CleanCLIP, indicating that we significantly reduced the ASR without compromising the model's performance and transferability. And the detailed information of these datasets are introduced as follows:

**CIFAR10 and CIFAR100**, introduced by Krizhevsky & Sutskever (2009), are small-scale color image datasets for image classification and object recognition tasks. CIFAR10 comprises 60,000 32x32 pixel color images in 10 classes, with 6,000 images per class. The dataset is split into 50,000 training images and 10,000 test images. CIFAR100 contains 100 classes grouped into 20 superclasses. Each class has 600 images, with 500 training images and 100 testing images. The superclasses organize the 100 classes in a semantic hierarchy.

**ImageNet1K**, often referred to as ILSVRC2012, is a subset of the ImageNet dataset Deng et al. (2009). It consists of approximately 1.28 million training images and 50,000 validation images, covering 1,000 classes. The dataset is widely used for large-scale image recognition tasks.

**DTD** (Describable Textures Dataset) Cimpoi et al. (2014) contains images of textures grouped into 47 categories based on a list of adjective-noun texture descriptions. Each category has 120 images, totaling 5,640 images. The dataset is designed for research in texture analysis and recognition.

**STL-10** dataset is designed for developing unsupervised feature learning, deep learning, and self-taught learning algorithms. It consists of 10 classes of unlabeled and labeled images, including aircraft, bird, car, cat, deer, dog, horse, monkey, ship, and truck. The images are 96x96 pixels in size.

**SVHN** (Street View House Numbers) is a real-world image dataset for digit recognition Netzer et al. (2011), derived from Google Street View images. It contains over 600,000 digit images coming from a variety of house numbers in Google Street View images. The dataset is partitioned into 73,257 training images, 26,032 test images, and 531,131 extra training images.

**Food-101** dataset consists of 101 food categories with 1,000 images per category Bossard et al. (2014). The images were collected from a large food image dataset available on the Internet. The dataset is designed for food recognition and related tasks.

**Oxford-IIIT Pet** Dataset Parkhi et al. (2012) is a large-scale dataset of pet images with fine-grained annotations. It consists of 37 pet categories with 200 images per category. The images exhibit large variations in scale, pose, and lighting. The dataset is designed for tasks such as fine-grained classification and segmentation.

**RenderedSST2** is designed to evaluate optical character recognition (OCR) capabilities in a sentiment analysis context Socher et al. (2013). It transforms textual sentiment labels into visual representations and comprises 60,000 32x32 rendered images of sentiment-labeled text (50,000 training images and 10,000 test images). The sentences belong to one of the 10 sentiment classes.

## D  DEFENSE PERFORMANCE ON EVA-CLIP

To validate the effectiveness of our method on semi-supervised tasks, we have extended our evaluation beyond the classic CLIP model discussed in the paper to include performance assessments on the EVA-CLIP as well (EVA-01-CLIP-B/16). The experiment settings of the attack process and finetuning defense are the same as our previous settings (Section 4.1). Detailed experimental results are presented in the following Table 4. This extended evaluation aims to further confirm the robustness and applicability of our defense strategy across different model architectures within the

Table 4: The defense performance of Top-k BA (%) and ASR (%) under the EVA-CLIP pertaining model (EVA-01-CLIP-B/16).

| Methods | | No defense | | CleanCLIP | | CleanerCLIP(**ours**) | |
|---|---|---|---|---|---|---|---|
| | | BA ↑ | ASR ↓ | BA ↑ | ASR ↓ | BA ↑ | ASR ↓ |
| BadNet | Top-1 | 61.85 | 91.76 | 53.67 | 3.89 | 53.35 | **0.93** |
| | Top-3 | 80.82 | 97.33 | 73.82 | 6.76 | 73.77 | **1.68** |
| | Top-5 | 88.23 | 98.12 | 81.94 | 11.27 | 82.03 | **3.44** |
| | Top-10 | 92.96 | 99.37 | 89.03 | 19.23 | 89.01 | **6.83** |
| BadCLIP | Top-1 | 60.87 | 99.35 | 52.16 | 80.23 | 52.13 | **17.92** |
| | Top-3 | 80.12 | 99.69 | 73.79 | 87.96 | 73.82 | **20.89** |
| | Top-5 | 88.44 | 99.78 | 81.23 | 90.65 | 81.25 | **24.44** |
| | Top-10 | 92.89 | 99.89 | 88.67 | 94.32 | 88.69 | **28.76** |

Table 5: The cross-dataset defense performance of Top-k BA (%) and ASR (%) against BadCLIP attack, fine-tuning with SBU-Captions dataset.

| Methods | | No defense | | CleanCLIP | | CleanerCLIP(**ours**) | |
|---|---|---|---|---|---|---|---|
| | | BA ↑ | ASR ↓ | BA ↑ | ASR ↓ | BA ↑ | ASR ↓ |
| BadCLIP | Top-1 | 58.60 | 98.81 | 49.50 | 87.24 | 49.53 | **15.96** |
| | Top-3 | 78.44 | 98.96 | 73.67 | 91.33 | 73.68 | **17.25** |
| | Top-5 | 84.51 | 99.14 | 81.09 | 93.62 | 81.11 | **24.66** |
| | Top-10 | 90.73 | 99.32 | 86.47 | 96.89 | 86.47 | **28.21** |

semi-supervised multi-modal contrastive learning framework. When we defend against BadNet attack, for Top-1 accuracy, CleanerCLIP achieves an ASR of 0.93%, a substantial improvement compared to CleanCLIP's 3.89% and No Defense's 91.76%. This indicates CleanerCLIP's capability to effectively mitigate the impact of the BadNet attack while maintaining a competitive accuracy of 53.35%. The performance is further enhanced in the Top-10 category, where CleanerCLIP records an ASR of 6.83%, compared to CleanCLIP's 19.23%, highlighting a robust defense that significantly lowers the vulnerability to this specific attack. Also, the effectiveness of CleanerCLIP is even more pronounced against BadCLIP, where it reduces the Top-1 ASR to 17.92%, vastly superior to CleanCLIP's 80.23%. This suggests that CleanerCLIP not only defends more effectively but also maintains a reasonable Top-1 accuracy of 52.13%. In the Top-10 evaluation, CleanerCLIP achieves an ASR of 28.76%, while CleanCLIP presents a significantly higher ASR of 94.32%, showcasing CleanerCLIP's robust capability to handle this more challenging attack scenario.

# E    CROSS-DATASET DEFENSE PERFORMANCE ON SBU-CAPTIONS

To further evaluate the effectiveness of our method, we have conducted performance assessments on the SBU-Captions dataset, extending beyond the previous evaluations on the classic CLIP model. The experimental settings for both the attack process and fine-tuning defense remain consistent with our prior configurations (Section 4.1). The detailed experimental results are presented in Table 5. This evaluation aims to provide additional insights into the robustness and adaptability of our defense strategy across diverse datasets within the semi-supervised multi-modal contrastive learning framework, reinforcing the efficacy of our approach in real-world scenarios. Notably, while the No Defense and CleanCLIP methods exhibit high ASR, our CleanerCLIP significantly reduces the ASR across all Top-k metrics. For instance, the Top-1 ASR drops dramatically from 87.24% (CleanCLIP) to 15.96%, showcasing a remarkable enhancement in defensive performance. Similarly, the Top-5 ASR improves from 93.62% (CleanCLIP) to 24.66%, and the Top-10 ASR decreases from 96.89% to 28.21%. These results highlight the effectiveness of our approach in mitigating the impact of adversarial attacks, confirming that CleanerCLIP not only maintains high accuracy but also significantly enhances robustness across different datasets, underscoring its unique and indispensable defense capabilities.

Table 6: The defense performance of ASR (%, ↓) with different text augmentations: RI (random insertion), RD (random deletion), RS (random swap), SP (synonym replacement), CFR-ori (counterfactual replacement with original captions) and CFR-sub (counterfactual replacement with pos-subcaptions, ours).

| Methods | | No defense | CleanCLIP | RI | RD | RS | SP | CFR-ori (ours) | CFR-sub (ours) |
|---|---|---|---|---|---|---|---|---|---|
| BadCLIP | Top-1 | 99.27 | 69.87 | 69.71 | 69.69 | 69.89 | 69.85 | **17.85** | **17.85** |
| | Top-3 | 99.66 | 82.72 | 82.57 | 82.54 | 82.73 | 82.74 | 20.21 | **20.19** |
| | Top-5 | 99.74 | 87.38 | 87.31 | 87.25 | 87.40 | 87.39 | 24.52 | **24.49** |
| | Top-10 | 99.83 | 91.96 | 91.73 | 91.83 | 91.91 | 91.95 | **28.09** | **28.09** |

# F  COMPARISON WITH EACH EDA TEXT AUGMENTATION USED IN CLEANCLIP

In this section, we compare the defense performance of CleanerCLIP against various Easy Data Augmentation (EDA) strategies employed in CleanCLIP. This comparison is crucial for evaluating the relative effectiveness of different augmentation techniques in mitigating the impact of backdoor attacks in multi-modal contrastive learning models.

As outlined in our motivation, we emphasize counterfactual replacement (CFR) as the core strategy within our method, which is designed to achieve semantic divergence at the text level. This divergence is critical for disrupting visual backdoor triggers within the feature space. To measure the degree of semantic change introduced by different text augmentation strategies, we utilize the metric "impact = 1 - cosine similarity" between the original and augmented captions. By applying five augmentation strategies to 1,000 samples from the CC3M dataset: random insertion (RI), random deletion (RD), random swap (RS), synonym replacement (SP), and counterfactual replacement (CFR), we computed the following impact values: RI (0.0434), RD (0.0516), RS (0.0396), SP (0.0247), and CFR (0.1015). Among these, CFR introduced the largest semantic change, demonstrating its potential to effectively break the correlation between text and image features, which is essential for neutralizing backdoor triggers.

In addition to measuring semantic change, we conducted a comprehensive evaluation of these text augmentation strategies under the BadCLIP attack scenario, where we overlaid each EDA strategy on CleanCLIP and measured the corresponding attack success rates (ASR). As shown in Table 6, CleanerCLIP, using counterfactual replacement, consistently outperformed other EDA strategies. For instance, under the Top-1 ASR evaluation, CFR-ori and CFR-sub both reduced ASR to 17.85%, compared to over 69% for the other EDA strategies (RI, RD, RS, SP). Similar improvements are observed across Top-3, Top-5, and Top-10 evaluations, where CFR-based augmentations demonstrate a significant drop in ASR, reaching as low as 20.19% (Top-3) and 28.09% (Top-10), highlighting the robustness of our approach.

Moreover, we investigated the effect of positive sub-caption selection (CFR-sub) versus original caption replacement (CFR-ori). The results show that both strategies yield nearly identical defense performance, confirming that the choice between positive sub-captions and original captions has minimal impact on overall defense effectiveness. Thus, for simplicity and practicality, we opted to use positive sub-captions (CFR-sub) in our experiments.

In summary, these evaluations underscore the superiority of counterfactual semantic replacement (CFR) over traditional EDA strategies. CleanerCLIP not only achieves the highest semantic divergence, but also provides the most effective defense against backdoor attacks across multiple metrics, demonstrating its unique and irreplaceable defense capabilities.

