# OpenReview forum: "CleanerCLIP: Fine-grained Counterfactual Semantic Augmentation for Backdoor Defense in Contrastive Learning"
_ICLR.cc/2025/Conference — ICLR 2025 Conference Withdrawn Submission_

### Official Review · Reviewer_4jUH · 2024-10-29

**Soundness:** 2
**Presentation:** 2
**Contribution:** 2
**Rating:** 3
**Confidence:** 4

**Summary:**

This paper proposes CleanerCLIP, a novel defense for fine-tuning multimodal contrastive models like CLIP against stealthy backdoor attacks. Unlike existing methods, CleanerCLIP uses fine-grained counterfactual text augmentation to disrupt connections between backdoor triggers and target features. It creates "positive" and "negative" subtexts, preserving clean semantics while randomly altering text components to break malicious links. This method significantly reduces ASR, especially against advanced attacks like BadCLIP, without compromising model accuracy on clean data, providing an efficient, lightweight solution for secure model fine-tuning.

**Strengths:**

1. The article provides an analysis of the shortcomings of current backdoor defense methods.
2. The article presents a method for creating Positive sub-captions and Negative sub-captions.

**Weaknesses:**

1. Section 3.1 only introduces the setting of the adversary and does not cover the setting of the defender.
2. Section 3.1 states that "the attacker can poison pre-trained MCL models by fine-tuning with carefully crafted dirty datasets," which is a strong assumption. To my knowledge, the first article [1] in this direction implemented a data poisoning setting without a separate fine-tuning phase on the poisoned data.
3. There is an overlap between the poisoning dataset and the fine-tuning dataset, which creates a simpler setting. When fine-tuning is performed on the overlapping data, the "backdoor connection" is naturally disrupted, even without any improvements being made.
4. The first paragraph of section 3.3 contains a **repeated paragraph**.
5. Where does the first term on the right side of Equation 4 come from? It hasn't been introduced before.
6. The second term on the right side of Equation 4 has not been introduced either.
7. How is the augmented data mentioned in the article being used? The introduction in section 3.3 is quite vague.
8. Due to the overlap between the poisoned data and the fine-tuning data in the experimental setup, there is a risk of data leakage.

[1] Carlini, N., & Terzis, A. (2021, October 6). Poisoning and Backdooring Contrastive Learning. International Conference on Learning Representations.

**Questions:**

Can you provide a concise and clear explanation of how the data augmentation method proposed in this article better disrupts the "backdoor connection"?
Can you provide experimental data to rule out the hypothesis that "the success of the method in this paper is mainly due to data augmentation covering the poisoned data"?

---

### Official Review · Reviewer_VBrG · 2024-11-01

**Soundness:** 3
**Presentation:** 2
**Contribution:** 2
**Rating:** 5
**Confidence:** 3

**Summary:**

This paper provides an effective finetuning backdoor defense mechanism for CLIP, namely *CleanerCLIP*. *CleanerCLIP* generates diverse counterfactual subtexts by replacing semantic components in texts, and uses them to disrupt the connection between the trigger and target feature during the defense fine-tuning. Besides, *CleanerCLIP* uses normal subtexts to maintain the performance of the fine-tuned models. Compared with the existing best finetuning backdoor defense strategy, *CleanCLIP*, *CleanerCLIP* performs better when the trigger is carefully optimized (e.g., close to the attack target instead of in different clusters) in the feature space (via BadCLIP). The authors evaluate the performance of *CleanerCLIP* against some existing backdoor attacks, and the results indicate that *CleanerCLIP* performs better than existing backdoor defense methods (e.g., *CleanerCLIP*).

**Strengths:**

1. The motivation and the insights of *CleanerCLIP* looks good and well presented, and the paper's introduction on related works (along with their pros and cons) is detailed.
2. The experimental results show the superior performance of *CleanerCLIP* over existing backdoor defense methods in CLIP. Ablation studys are comprehensive.
3. In all, the paper is not hard to read.

**Weaknesses:**

1. It is good to explore different Top-K metrics between *CleanerCLIP* and other baselines, and it can be better for the authors to consider more attacker's target (not just banana). As mentioned by the authors in Line 235->Line 240, *CleanerCLIP* uses various ways to craft negative sub-captions due to the defender "*cannot precisely know which entity, attribute, or relationship the adversary might exploit*,"; thus, covering more attacker's targets (e.g., other subjects such as animals, or verbs such as different behaviors) in the evaluation can strengthen this claim.

2. Some writing/presenting details in the paper can be further polished. For example:

	(a). There seems to be repeated statements (Line 250->Line 256 (Previous defense efforts ...) has very similar meaning with Line 256->Line 259 (Previous defense strategies ...)) in Sec 3.3. The authors can double check this paragraph.

	(b). The format of Figure 3 and Figure 4 seems different. The plots in Figure 4 look like pdf-format (the words can be selected) while those in Figure 3 look like pnd/jpg-format (a little blurred and the words cannot be selected). It is better to align the format of all plots in the paper.

	(c). The label "TA-C" in Figure3 seems not well introduced in the paper (though we know it means *CleanerCLIP*), because I cannot find it (by searching TA-C in the pdf) in the main text. I think the authors can explain it in the figure caption or somewhere in the text.

	(d). The authors can describe the hyperparameters of the baseline (e.g., the blending ratio of Blended) in details, or it can be better to show the visual effect of all involved backdoor triggers.

	(e). There may be some typo errors in Eq. (1) and (2) in the main text. For example: in (1)'s denominator, $\sum_{k=1}^{K} \exp \left( \langle z_i^n, z_i^I \rangle / t_n \right)$ seems not involve "k" in its items $\exp \left( \langle z_i^n, z_i^I \rangle / t_n \right)$ (maybe just multiple $\exp \left( \langle z_i^n, z_i^I \rangle / t_n \right)$ with $K$?). And it seems that Eq. (1) and (2) are the same thing, the only difference is the order of two items in "<>" (for example: first item in the denominator: $\sum_{j=1}^{K} \exp \left( \langle z_i^I, z_j^p \rangle / t_p \right)$ in Eq. (1) and $\sum_{j=1}^{K} \exp \left( \langle z_j^p, z_i^I \rangle / t_p \right)$ in Eq. (2)). Please double check these equations.

**Questions:**

1. How about practical feature distribution (including triggers, targets, and others) before/after *CleanerCLIP* corresponding to Figure 1 (d)? It can be better for the authors to show the practical effect of *CleanerCLIP* in the feature space, just like *CLEANCLIP*'s[1] Figure 2.
2. During the training of *CleanerCLIP*, $z_i^I$ is to be far away from other data's positive sub-captions $z_j^p$ ($j \in [1, K]$) and self's negative sub-caption $z_i^n$ (according to Eq. (1)). In this case, I think $z_j^p$ ($j \in [1, K]$) serves as another group of negative sub-captions (also with distinct semantic information of $z_i^I$). If so, why adding self's negative sub-caption into the finetuning can bring about so apparent backdoor mitigation performance? Is it because $z_i^I$ has more diverse and sparse feature distribution than $z_j^p$ ($j \in [1, K])$? If so, I'm kind of curious about whether just performing similar augmentation on $z_j^p$ ($j \in [1, K])$ can mitigate backdoor effect during the finetuning. Never mind if you have no time to do related experiments, but I'd like to learn about your insights on my querys (and if there is something wrong in my query, feel free to point out).
3. The BA metric can also be put in Figure 4 (b)(c) to see the trade off between it with ASR when adjusting hyperparameters of *CleanerCLIP*. (Besides, I think *BadCLIP* maybe a better defense target based on the advantage of the proposed method.)
4. See weaknesses.

[1] "CleanCLIP: Mitigating Data Poisoning Attacks in Multimodal Contrastive Learning" ICCV 2023 Bansal et. al.

---

### Official Review · Reviewer_92Xh · 2024-11-04

**Soundness:** 3
**Presentation:** 2
**Contribution:** 2
**Rating:** 6
**Confidence:** 2

**Summary:**

The motivation of this paper is to address the limitations of the existing backdoor defense method (CleanCLIP) of CLIP, which is less effective against certain backdoor attacks (BadCLIP). Specifically, CleanerCLIP first constructs a set of positive and negative sub-captions and then uses contrastive learning to disrupt the hidden trigger-target mapping. Experimental results show that this method successfully defends against the "BadCLIP" attack, whereas CleanCLIP does not.

**Strengths:**

1.	Compared to the baselines, this method is generally effective against the backdoor attacks in Tab. 1, with promising experimental results.

2.	Extensive ablation study.

**Weaknesses:**

1.	Poor writing: incorrect citation formatting makes the Intro difficult to read. E.g. "Notable approaches include CleanCLIP Bansal
et al. (2023) and RoCLIP Yang et al. (2024a)".

2.	Leak of quantitative analysis of the impact of Sub-caption and Neg-caption generation strategies on the results. Intuitively, Sub-caption and Neg-caption generation could have a significant effect on the final results.

3.	CleanerCLIP may only be applicable to multimodal backdoors triggered by images, which could limit its range of applications (The authors acknowledged this in the limitation, which I appreciate.). Additionally, this paper seems somewhat focused on addressing weaknesses in CleanCLIP [1], raising concerns that it might offer only incremental technical contributions.

[1] Bansal, Hritik, et al. "Cleanclip: Mitigating data poisoning attacks in multimodal contrastive learning." Proceedings of the IEEE/CVF International Conference on Computer Vision. 2023.

**Questions:**

In Figure 4(c), a higher value of "Cleaner-sample nums" leads to a lower ASR, which is easy to understand. On the other hand, does a higher value of "Cleaner-sample nums" also degrade the model's utility?

---

### Official Review · Reviewer_EkY6 · 2024-11-04

**Soundness:** 2
**Presentation:** 2
**Contribution:** 2
**Rating:** 3
**Confidence:** 4

**Summary:**

This paper introduces a fine-tuning-based backdoor defense, CleanerCLIP, targeting, as claimed, more stealthy backdoor attacks. CleanerCLIP introduces a sub-text strategy where positive sub-caption and negative sub-caption are generated and later used for fine-tuning. In particular, positive sub-captions are generated using a pre-defined template that preserves the relationships between subjects and objects, and negative sub-captions are generated by replacing adjectives, relational verbs, and subject/nouns with the help of WordNet and ChatGPT. Similarities between sample pairs are used for fine-tuning.

**Strengths:**

This work recognizes the current issue of recent backdoor defenses against backdoor attacks, claiming that the self-supervision strength in CleanCLIP is insufficient against backdoors that leverage existing clean features. This hypothesis is interesting to explore further. Detailed background details are introduced to provide users with a complete background. Extensive experiments are provided to evaluate the proposed defense. However, some claims need to be further clarified.

**Weaknesses:**

- About the clean feature exploitation. The authors claimed in Line 257 that "self-supervision strength in CleanCLIP is insufficient against the meticulously crafted triggers that leverage existing clean features without creating additional toxic feature clusters." However, this point is not underpinned comprehensively. I anticipate this is a new finding from this paper, so further clarifications could be provided to support this claim since this is the motivation behind the defense design. In particular, why this is a limitation instead of an advantage? What is the connection between BadCLIP and this observation? Why could the proposed strategy be a good fix for this observation, let alone the provided experimental analysis? I expect that the advantage of the CleanerCLIP will be further strengthened after clarifying this point.

- Another related question is about the stealthy backdoor techniques. The definition of stealthy has been mentioned several times without a complete definition. Given the importance of the observation that current backdoor attacks are ineffective against more stealthy backdoor techniques, it would be great if the authors could clarify the definition of stealthy for backdoor attacks. For instance, does it indicate the exploitation of the clean feature instead of creating backdoor features?

- Incremental improvement. Looking at table 1, the performance of ClearnerCLIP is not substantially stronger than CleanCLIP against BadNets and Blended. It would be great if the authors could further clarify this observation. Especially please explain the performance difference between BadCLIP and Blended/BadNet. Does it indicate the clean feature exploitation observation does not hold for Blended and Badnet?

**Questions:**

Please clarify the clean feature exploitation, the definition of stealthy, and the incremental improvement.

---

### Note · Authors · 2024-11-13

I have read and agree with the venue's withdrawal policy on behalf of myself and my co-authors.